# Extensive Tandem Duplication Events Drive the Expansion of the C1q-Domain-Containing Gene Family in Bivalves

**DOI:** 10.3390/md17100583

**Published:** 2019-10-14

**Authors:** Marco Gerdol, Samuele Greco, Alberto Pallavicini

**Affiliations:** 1Department of Life Sciences, University of Trieste, 34127 Trieste, Italy; SAMUELE.GRECO@phd.units.it (S.G.); pallavic@units.it (A.P.); 2National Institute of Oceanography and Applied Geophysics, 34151 Trieste, Italy

**Keywords:** innate immunity, lectins, complement system, C1q, bivalve mollusks, tandem duplication, pattern recognition receptors

## Abstract

C1q-domain-containing (C1qDC) proteins are rapidly emerging as key players in the innate immune response of bivalve mollusks. Growing experimental evidence suggests that these highly abundant secretory proteins are involved in the recognition of microbe-associated molecular patterns, serving as lectin-like molecules in the bivalve proto-complement system. While a large amount of functional data concerning the binding specificity of the globular head C1q domain and on the regulation of these molecules in response to infection are quickly accumulating, the genetic mechanisms that have led to the extraordinary lineage-specific expansion of the C1qDC gene family in bivalves are still largely unknown. The analysis of the chromosome-scale genome assembly of the Eastern oyster *Crassostrea virginica* revealed that the 476 oyster C1qDC genes, far from being uniformly distributed along the genome, are located in large clusters of tandemly duplicated paralogs, mostly found on chromosomes 7 and 8. Our observations point out that the evolutionary process behind the development of a large arsenal of C1qDC lectin-like molecules in marine bivalves is still ongoing and likely based on an unequal crossing over.

## 1. Introduction

A growing body of evidence supports the idea that a proto-complement system, composed of C3, factor B and complement receptors, has an ancient origin in the animal tree of life [1]. Although several lineage-specific gene losses and acquisitions are likely to have reshaped the organization of the proto-complement system during its evolution, leading to its complete loss in some major extant taxa (e.g., insects), the activation of this primary defense system often relies on the recognition of microbe-associated molecular patterns (MAMPs) by different types of soluble lectin-like molecules. In vertebrate animals, two different types of pattern recognition receptors (PRRs) are involved in the lectin pathway of the complement system: mannan-binding lectins (MBLs) and ficolins. These proteins can recognize carbohydrate moieties associated with pathogens, triggering the activation of the complement proteolytic cascade, and eventually leading to the opsonization and killing of invading microbes.

The vertebrate complement component C1q is a connecting link between innate and adaptive immunity, as it enables complement activation through the recognition of antigen-complexed immunoglobulins, in the second arm of the complement system—the so-called classical pathway. Due to the absence of immunoglobulins, a proper classical pathway does not exist in invertebrates, making the activation of the complement system only possible in response to MAMP recognition by lectins, to the spontaneous hydrolysis of C3 (i.e., the alternative pathway), or to the direct binding of MAMPs by C1q, which is also observed in vertebrates [2]. In addition to the three monomeric units of the vertebrate C1q complex, the globular head C1q domain is also found in a number of different proteins with non-complement related functions, such as cerebellin, adiponectin, emilin, multimerin and others [3], defining the C1q-domain-containing (C1qDC) protein family and highlighting the high versatility of this structural scaffold [2].

In spite of the absence of an adaptive immune system, a large number of C1qDC proteins are encoded by the genomes of numerous metazoans which lack an adaptive immune system [4]. Due to the presence of several hundred C1qDC genes, bivalve mollusks most certainly emerge as the prime example of an animal group in which C1qDC proteins have met significant evolutionary success. For example, in stark contrast with most other protostomes, including gastropod and cephalopod mollusks, which only produce less than a dozen different C1qDC proteins [5,6], the genome of the Pacific oyster *Crassostrea gigas* harbors 337 C1qDC genes. Multiple transcriptome [7,8] and genome sequencing efforts have confirmed that C1qDC genes contribute to 0.5–1.5% of the entire repertoire of protein-coding genes of most bivalve species (e.g., 296 genes in *Pinctada fucata* [9], 445 in *Modiolus philippinarum* [10], 554 in *Saccostrea glomerata* [11] and over 1200 in *Ruditapes philipinarum* [12]). Curiously, this massive gene family expansion has been inferred to have occurred quite recently in bivalve evolution, since it only targeted all Pteriomorphia and Heterodonta, regardless of the environmental niche, but not the two remaining basal classes of Palaeoheterodonta and Protobranchia [13]. 

While it is still unclear whether all bivalve C1qDC proteins are involved in immune recognition [13], functional studies indicate that many of them play an important role as lectin-like molecules. The binding properties of the C1q domain enable the recognition of a broad range of MAMPs, such as peptidoglycan (PGN) and lipopolysaccharide (LPS)—the major components of Gram-positive and negative bacterial cell walls respectively—but also of other sugars associated with invading microbes, such as mannan [14,15], beta-1-3-glucan and yeast-glucan [16,17,18]. The impressive molecular diversification of bivalve C1qDC proteins has been hypothesized to be linked with a parallel functional specialization [16], which may further extend the range of potentially recognized MAMPs [19,20].

Bivalve C1qDC proteins are expressed in different tissues [13] and, upon secretion in the extracellular environment, they might be released in the hemolymph [7], in the extrapallial fluid [21] or in the mucus that covers the gills [13], offering a first line of defense against invading microorganisms in different body districts. The recognition of MAMPs by bivalve C1qDC proteins, which is probably aided by additional humoral factors, promotes the agglutination of bacterial cells [15,22], also triggering the migration and phagocytic activity of hemocytes [22,23,24], which clearly indicate an opsin-like function for these important soluble PRRs.

In vertebrates, the activation of the complement proteolytic cascade by C1q is effected by the presence a collagen tail, which also enables trimerization and the formation of a typical bouquet structure and defines the C1q-like type I domain architecture [20]. However, collagen tails are extremely rare in bivalves, which seem to either rely on a functionally analogous coiled-coil region for the assembly of oligomeric complexes (C1q-like type II proteins) and often completely lack N-terminal extensions (sghC1q proteins) [13].

Although several functional aspects remain to be fully investigated, the past decade has witnessed significant progress in the study of bivalve C1qDC proteins. Although these reports have contributed to a better elucidation of their functional significance in the context of immune response, the unavailability of high-quality genome assemblies has so far prevented the study of the genetic and molecular mechanisms that have led to the generation of several hundred C1qDC genes in this class of aquatic filter-feeding metazoans. Here, through the analysis of a high-quality chromosome-scale genome assembly [25], we investigate the genomic organization of the 476 C1qDC genes found in the Eastern oyster *Crassostrea virginica*, revealing a highly inhomogeneous distribution across chromosomes and a still-ongoing gene family expansion process which is mostly based on tandem gene duplication and episodic positive selection acting on newly generated gene copies. This process may have a significant impact of the functional diversification of these lectin-like molecules, resulting in the extension of the range of recognized MAMPs.

## 2. Results

### 2.1. The Repertoire of the Eastern Oyster C1qDC Genes

In total, 476 C1qDC genes were identified in the genome of *C. virginica* (detailed in Appendix A ). This number is in line with the previous report of 337 C1qDC genes in the congeneric species *C. gigas* [13], whose genome is slightly smaller (558 Mb vs. 685 Mb) [26], and similar to other Pteriomorphia [9,10,11].

Following the classification scheme previously proposed in another publication [13], oyster C1qDC proteins were labeled as follows: (i) sghC1qDC proteins, i.e., proteins containing a signal peptide, immediately followed by the C1q domain; (ii) sC1q-like type I proteins, i.e., secreted proteins containing a collagen tail before the C1q domain; (iii) sC1q-like type II proteins, i.e., secreted proteins containing a coiled-coil tail before the C1q domain; (iv) smultiC1q, i.e., secreted proteins containing multiple C1q domains; (v) other/uncertain, i.e., proteins with different domain architectures, or those resulting from likely incomplete annotation.

As in the case of the Pacific oyster, the majority of the C1qDCgenes (262, 55%) belonged to the sC1q-like type II category. SghC1q proteins were the second most abundant type, with 111 genes (23%); 21 genes encoded proteins with multiple C1q domains (three in most cases), which may or may not include a coiled-coil region (Table 1). No C1q-like type I protein was found in the Eastern oyster, confirming the observation that the association between the C1q domain and *N*-terminal collagen regions, typical of vertebrates, rarely occurs in bivalves [7,13].

A total of 79 genes were classified in the “other/uncertain” category. While in part these may correspond to truncated genes resulting from incorrect computational prediction (as previously shown for many C1qDC genes in *C. gigas* [13]), some may correspond to bona fide cytosolic or membrane-bound C1qDC proteins, which have been previously identified in other bivalve species [7,13]. An interesting subgroup of unusual C1qDC proteins was found to contain a d-galactoside/l-rhamnose-binding SUEL domain (acronym for sea urchin egg lectin), which is typically found in a number of lectins produced by marine echinoderms [27,28,29]. Although three sSUEL/C1q genes had been previously reported in the Pacific oyster, their number in the Eastern oyster (16) largely exceeds that of the congeneric species [13].

### 2.2. Chromosomal Distribution of Oyster C1qDC Genes

Far from being uniformly split among chromosomes, oyster C1qDC genes displayed a highly skewed distribution, with more than 50% genes being located on chromosomes 7 and 8. Indeed, while these two chromosomes contained 123 C1qDC genes each, others encoded as few as 12 C1qDC genes (i.e., chromosomes 4 and 10). This reflected a very uneven C1qDC gene density, ranging from 0.17 genes/Mb (chromosome 5) to 2.13 genes/Mb (chromosome 7), with the average genomic C1qDC gene density standing at 0.70 genes/Mb (Figure 1).

C1qDC gene density was also largely non-homogeneous along chromosomes, as the genes were often placed in packed clusters, which often included more than a dozen units. This resulted in local peaks of very high gene density (i.e., >20 C1qDC genes/Mb in some regions of chromosome 7 and 8), as opposed to very large regions (>10 Mb) completely devoid of C1qDC genes (e.g., chromosome 3, 5 and 10) (Figure 1). Overall, 1.38% of the genes encoded by the *C. virginica* genome pertain to the C1qDC gene family. The relative abundance of C1qDC genes was however much higher for the two aforementioned chromosomes, standing at 4.78% and 3.16% for chromosome 7 and 8, respectively.

At the same time, a complete view of the localization of C1qDC genes in the genome (Figure 1) clearly shows that the different types of C1qDC genes were not evenly distributed. Type II C1q-like genes were the most abundant in all chromosomes, with the exception of chromosome 8, where the sghC1q type was the predominant one (Table 1). The high abundance of sghC1qDC genes in chromosome 8 was particularly notable, as 63 out of the 111 C1qDC genes of this type (57%) were found in this chromosome. Similarly, the majority of smultiC1qDC genes (14 out of 21) were found on chromosome 8, located in a dense cluster (Figure 1). On the other hand, most sSUELC1q genes (10 out of 16) were located on chromosome 2. 

### 2.3. Most Oyster C1qDC Genes Are Tandemly Duplicated

The organization of oyster C1qDC genes in monotypic gene clusters (Figure 1) clearly suggest that these sequences are paralogous, which may have been generated by tandem gene duplication. We analyzed this evolutionary aspect in more depth by (i) inspecting the pairwise sequence similarity of the encoded proteins, and (ii) investigating their relative position. The clustering approach evidenced that 421 C1qDC proteins (88.44%) were grouped in clusters of at least two putative paralogs, and that 172 genes (36.13%) were grouped in large clusters (i.e., including ≥10 putative paralogous genes).

The C1qDC genes included in such large clusters were, for the most part, both evolutionarily closely related and spatially close to each other, as exemplified by one of the largest clusters identified in chromosome 7 (Figure 2a), which included 34 C1qDC genes, mostly of the C1q-like type II group. The phylogenetic analysis revealed that, of all C1qDC genes encoded by chromosome 7, these 34 genes created a well-supported monophyletic group (97% posterior probability), and they were organized in two distinct gene subclusters, containing 6 and 28 genes, respectively. The two subclusters were found to be separated by ~1.8 Mb sequence (Figure 2a) and, for the most part, included tandemly duplicated genes encoded on the same strand.

The investigation carried out at the whole-genome scale confirmed the widespread occurrence of tandem duplications, which impacted 297 C1qDC genes (62.39%). In total, 84 genes (17.65%) were proximally duplicated (i.e., they were not directly flanked by C1qDC genes, but placed within 100 Kb distance from the closest one), and thus were the likely product of similar unequal crossing over processes [30] (Figure 2b). As previously mentioned, only a tiny fraction of the Eastern oyster C1qDC genes (i.e., <12%) were single-copy (i.e., they encoded proteins sharing <50% sequence similarity with other C1qDC proteins) and, in accordance with this observation, the occurrence of dispersed C1qDC genes (95 genes, 19.96%) was only slightly higher, showing the low prevalence of gene duplications driven by the activity of transposable elements. Chromosomes 7 and 8—i.e., the two chromosomes with the highest C1qDC gene density—were also those containing the highest proportion of tandemly or proximally duplicated genes (91.06% and 86.99%, respectively), confirming the fundamental importance of these processes in the bivalve C1qDC gene family expansion.

### 2.4. The Gene Duplication Process Is Still Ongoing and Is Paired with Diversifying Selection

Multiple lines of evidence suggest that the development of a very large repertoire of C1qDC genes in the Eastern oyster is the product of an evolutionary process that is still presently ongoing. Evidence in support of the very recent origin of paralogous C1qDC gene copies by tandem duplication is provided, for example, by the existence of eight nearly-identical genes, which encode proteins sharing 100% sequence identity at the amino acid level. In particular, three paralogous sghC1qDC genes located on chromosome 7 (i.e., LOC111103100, LOC111103101 and LOC111103102) share 100% sequence identity at the nucleotide level within the coding sequence, and only display small intronic indels due to the presence of microsatellites. A total of 65 C1qDC genes (13.63%) encode proteins with >95% sequence identity with the closest paralog, and this number rises to 89 genes (18.66%) for a sequence similarity threshold equal to 90%.

An additional example of this evolutionary process is given by a group of six highly similar paralogous sC1q-like type 2 genes found in chromosome 8 (i.e., LOC111105959, LOC111105960, LOC111105962, LOC111106427, LOC111106428 and LOC111109313), which all encode proteins of very similar length (249–251 aa) and share >90% sequence identity (Figure 3a). The six genes retain the same gene architecture, with conserved splicing donor and acceptor sites. Similarly, they display a nearly identical size of exons (with the only two exceptions of a 3nt-long in-frame deletion in LOC111105962 exon 1, and a 1nt-long deletion at the end of the coding sequence of LOC1111105960, which resulted in the acquisition of an additional codon at the 3’ end of the ORF). As expected, the intron was subject to more relaxed selective constraints compared with the two exons, displaying a higher evolutionary rate, as evidenced by the higher number of indels and SNPs (Figure 3a).

BUSTED revealed strong evidence (*p*-value < 0.05) of episodic positive selection in the six largest clusters of paralogous C1qDC genes identified with a genome-wide analysis, suggesting that the gene duplication process is rapidly followed by molecular (and possibly functional) diversification through positive selection acting on a limited number of specific sites. In contrast with these hypervariable sites, the vast majority of the sites included in the globular C1q domain were found to evolve under negative selection, likely due to selective constraints linked with the maintenance of the typical 10-strand jelly-roll fold structure of the C1q domain itself, as well as of the contact surfaces between subunits required for oligomerization. The number of sites subject to significant diversifying selection, and their statistical support, largely varied from one cluster to the other. The four clusters that included C1q-like type 2 genes (i.e., cluster 10: 18 genes, cluster 22: 26 genes, cluster 81: 15 genes, and cluster 88: 19 genes) displayed between seven and 11 positively selected sites; i.e., roughly 5–10% of the sites included in the globular C1q head. The two clusters comprising sequences of different types (i.e., cluster 25: 16 sSUELC1q genes; cluster 102: 21 sghC1q genes) displayed a somewhat lower number of positively selected sites compared to the four aforementioned cases (four and five, respectively) (Figure 3b).

## 3. Discussion

In eukaryotes, small-scale gene duplication (i.e., independent from whole-genome duplication) can occur through a number of different mechanisms [31], which may or may not include the activity of transposable elements. Tandem gene duplications result in the creation of two adjacent paralogous genes, which are usually separated by a few Kb of intergenic sequence. Proximal gene duplications similarly result in the creation of two paralogous genes, which are placed at slightly longer distances compared with tandem duplications, and usually separated from each other by one or more genes. Both types of duplication are generally thought to arise from an unequal crossing over; i.e., from the misalignment of homologous regions of sister chromatids during meiosis [30]. An alternative process behind the creation of paralogous gene copies depends on the activity of transposable elements (TEs), such as DNA transposons [32] or retrotransposons [33]. In this case, new gene copies may either retain the original exon/intron architecture and regulatory features (DNA transposons), or completely lack introns, being detached from the promoter region of the original paralogous copy (retrotransposons). Even though the activity of TEs is sometimes spatially localized [34], in most cases, the new gene copies are dispersed; i.e., they are placed in genomic regions distant from their paralogs, often in different chromosomes.

Our investigation demonstrates that the predominant evolutionary process behind the massive expansion of the C1qDC gene family in bivalves is tandem gene duplication. This conclusion is supported by the observation that 62.39% of paralogous C1qDC genes are tandemly duplicated and that an additional 17.65% are proximally duplicated (Figure 2b). Hence, the widespread presence of C1qDC gene clusters in the Eastern oyster genome suggests that an unequal crossing over might be considered as the primary driving force behind this gene family expansion event. Our study further suggests that this process is still actively ongoing, since numerous identical or nearly-identical gene copies, inferred to have a very recent evolutionary origin, are present in dense gene clusters, which shows a highly inhomogeneous distribution along the genome.

However, this remarkable gene family expansion, with the extensive retention of newly generated gene copies, requires some additional explanation, as it seemingly contradicts the observation that the majority of duplicated genes with redundant function evolve under more relaxed selective constrains, being more prone to loss-of-function mutations, pseudogenization and consequent loss [35,36]. Over the years, multiple and partially overlapping theoretical models have been proposed to explain the fixation of duplicated genes in populations, which depend on a complex combination between the evolutionary dynamics of a given gene family and functional properties of the encoded proteins [37,38]. The retention of hundreds of C1qDc genes in the Eastern oyster, as well as in other bivalve species [5,7,13], most certainly indicates that the generation of tandemly and—in a minor way—proximally duplicated C1qDC genes represents an important evolutionary advantage in these filter-feeding organisms. Based on the functional information collected so far in oysters, scallops, mussels, clams and other bivalves, this advantage could arise from the neofunctionalization or subfunctionalization of new gene copies [39], which may acquire the ability to recognize additional MAMPs, extending the range of pathogenic microorganisms recognized, through the mutation of key sites involved in glycan recognition [16].

One key aspect that remained to be investigated was whether such molecular diversification is the result of relaxed selective constraints on redundant gene copies, or of positive selection acting on selected sites potentially involved in MAMP-binding, which could significantly affect the lectin-like function of oyster C1qDC proteins. The analysis of the six largest clusters of paralogous genes found in the *C. virginica* genome revealed highly supported traces of episodic positive selection, suggesting that duplicated gene copies undergo rapid molecular, and possibly functional, diversification. This process only seemed to act on a limited number of localized residues (4–11 per cluster, Figure 3b), which were in part shared by the different clusters and displayed a higher frequency in the hypervariable and gap-rich C-terminal part of the C1q domain (Figure 3b). On the other hand, the vast majority of the sites included in the C1q domain was subject to strong purifying selection, likely due to structural constraints. Our current knowledge of the sites involved in MAMP recognition in bivalve C1qDC proteins is virtually non-existent, and most of the information available concerning the residues involved in C1q ligand binding derives from very distantly related model organisms, such as mouse and human [40,41]. Consequently, the possibility to infer whether the position of positively selected sites in the 3D structure of the globular head C1q domain is paired with a functional specialization and to the acquisition of the ability to recognize novel ligands in tandemly duplicated C1qDC genes falls beyond our current reach.

Another interesting aspect revealed by this study was the complete lack of C1q-like type I proteins in *C. virginica*. In light of the recent report of the presence of this domain architecture in early-branching metazoans [42], its absence in the Eastern oyster suggests that bivalves do not necessarily require collagen tails for the activation of the proto-complement system. The presence of a very few non-orthologous C1q-like type I genes in other bivalve species [7,13] further reinforces the idea that alternative N-terminal structures (i.e., coiled-coil regions) are the predominant structure used for C1qDC protein oligomerization and the subsequent activation of the complement proteolytic cascade. The finding that 16 sSUEL/C1q genes were present in the Eastern oyster genome was another relevant result, since this domain combination, unique to Bivalvia, is highly reminiscent of the recently reported case of C1q-related proteins (QREPs) found in a few Caenogastropoda and Heterobranchia gastropod species [42]. As with sSUEL/C1q, these unusual proteins combine the C-terminal globular C1q domain with N-terminal domains with marked binding properties (i.e., two immunoglobulin-like domains). Altogether, these findings stimulate further research towards the functional characterization of these potentially multifunctional lectin-like proteins.

In summary, this study provides, for the first time, evidence supporting the important role of tandem gene duplication and an unequal crossing over in the expansion and molecular evolution of the C1qDC gene family in bivalves. Based on preliminary results, this rapid and still-ongoing process is likely paired with episodic diversifying selection, which only acts on a limited number of spatially localized residues, whose functional significance in MAMP binding should be investigated in depth in the future. Evolutionary processes analogous to those we have described for the Eastern oyster C1qDC genes may have targeted several other gene families involved in immune response, either as receptors or as effectors, that have similarly reportedly undergone massive lineage-specific expansion and molecular diversification [5,43,44].

## 4. Materials and Methods 

### 4.1. Identification of C1qDC Genes

The annotated chromosome-scale nuclear genome assembly v.3.0 of *Crassostrea virginica* (Gmelin, 1791) [25] was downloaded from the GenBank repository (GCA_002022765.4). Protein-coding genes were virtually translated and the resulting amino acid sequences were screened for the presence of one or more C1q domains with HMMER v.3.2.1 [45], based on a 0.05 e-value threshold, using the PFAM profile of the C1q domain (PF00386) as a query. Positive hits were further refined by removing sequences with partial domains, taking into account the possibility of incorrect annotations.

All C1qDC proteins were further characterized as follows: the presence of *N*-terminal signal peptides and transmembrane regions was inspected with Phobius [46]; additional conserved protein domains were predicted with InterProScan v.5 [47] and coiled-coil regions were predicted with COILS [48], with a window length of 14, 21 and 28 amino acids, based on a probability threshold > 0.5. Protein nomenclature followed the scheme proposed in a previous publication [13] (see Section 2.1 for details).

### 4.2. Sequence Analysis of C1qDC Genes

The coordinates of each C1qDC gene were obtained from the genome GFF (General Feature Format) annotation file, and their positions were indicated on chromosomes, drawn to scale. The four gene categories indicated in Section 4.1 (with the exception of type I C1q-like genes, as no sequence of this type was identified) were indicated with different colors. The C1qDC gene density per Mb of genomic DNA sequence was computed using a 1 Mb sequence window length, and represented as a heat map on the side of each chromosome.

The amino acid sequences were clustered by similarity using CD-HIT v.4.8.1 [49], with different sequence identity thresholds to identify the products of putative gene duplication events. Sequences sharing > 50% pairwise identify were considered to be part of paralogous gene clusters and further classified as (i) tandemly duplicated genes, if they were flanked by at least one paralogous C1qDC genes, either at the 5’ or 3’ side; (ii) proximally duplicated genes, if the flanking genes were not pertaining to the C1qDC family, but the closest paralogous C1qDC was located within 100 Kb of distance; and (iii) dispersed genes, if the closest C1qDC gene was not located within 100 Kb of distance.

The evolutionary history of the C1qDC genes encoded by chromosome 7 (i.e., the oyster chromosome displaying the highest C1qDC gene density) was further investigated by generating a multiple sequence alignment (MSA) of all the encoded amino acid sequences. The MSA file, obtained with MUSCLE v.6.0 [50], was trimmed to only include the C1q domain; i.e., the region shared by all proteins. For smultiC1q proteins, each domain was separately added to the MSA. ModelTest-NG v.0.1.5 [51] estimated the WAG (Whelan And Goldman) model of molecular evolution [52], with a proportion of invariable sites and a gamma-shaped rate of variation across sites (WAG+G+I), as the best-fitting for the dataset analyzed. Phylogenetic inference was carried out with MrBayes v.3.2.7a [53], running two parallel MCMC analyses with four chains each. The convergence of all estimated parameters was assessed with Tracer v.1.7.1 [54], based on the reaching of an effective sample size > 200; i.e., with 400,000 generations. The resulting phylogenetic tree was represented as a 50% majority rule consensus tree (i.e., nodes supported by posterior probability values <50% were collapsed).

### 4.3. Positive Selection Analysis

The six largest clusters of paralogous C1qDC genes, identified as described in Section 2.2, all containing 15 genes or more, were analyzed to detect the signatures of positive and negative selection as follows. First, the nucleotide sequences of the open reading frames were aligned with MEGA X [55], using MUSCLE [50], preserving the integrity of codon triplets. The obtained MSA files were trimmed to remove positions containing gaps in the alignment, as well as regions not included in the globular C1q domain. The resulting processed MSA files were processed with BUSTED [56] to detect domain-wise signatures of episodic positive selection, based on a *p*-value threshold < 0.05. The position of sites subject to positive selection was predicted with FUBAR, SLAC, FEL and MEME [57,58,59], based on a *p*-value threshold < 0.05 (or posterior probability > 0.95). All these tools were implemented with the use of the online Datamonkey 2.0 platform (https://datamonkey.org) [60]. Sites evolving under positive selection were categorized either as strongly supported (i.e., detected with at least two different methods) or as moderately supported (i.e., detected with just one method). Sites subject to positive selection were graphically represented in the alignment between the consensus sequences of the six selected clusters, using the experimentally determined structure of the chain A of the human C1q (PDB: 1PK6) [19] as a reference. The multiple sequence alignment was implemented with structural information using Expresso [61].

## Figures and Tables

**Figure 1 marinedrugs-17-00583-f001:**
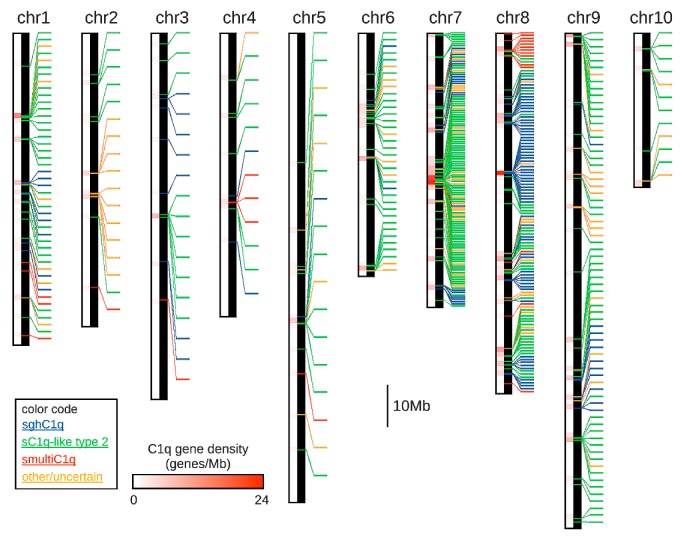
Chromosomal distribution of C1qDC genes in the *Crassostrea virginica* genome. The three main types of C1qDC genes (sghC1q, sC1q-like type 2, and smultiC1q) are indicated with different colors. C1qDC genes encoding proteins with a diverse domain organization, or of uncertain classification, were placed in the “other/uncertain” category. Chromosomes are drawn in scale.

**Figure 2 marinedrugs-17-00583-f002:**
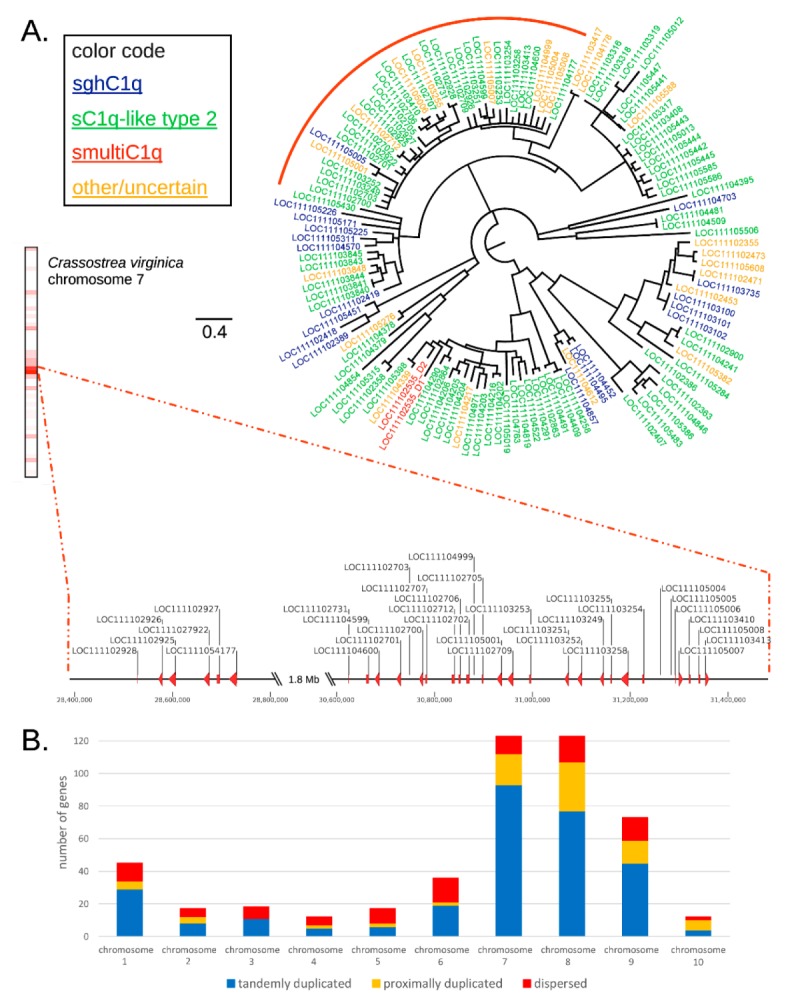
(**A**) Bayesian phylogenetic tree of the C1qDC proteins encoded by *Crassostrea virginica* chromosome 7. The three main types of C1qDC genes (sghC1q, sC1q-like type 2 and smultiC1q) are indicated with different colors. The location of the C1qDC genes pertaining to the monophyletic group indicated with a red circle is shown in detail in the zoomed-in chromosomal region. Nodes supported by posterior probability values <50% were collapsed. (**B**) Number of tandemly duplicated, proximally duplicated and dispersed C1qDC genes in each *C. virginica* chromosome.

**Figure 3 marinedrugs-17-00583-f003:**
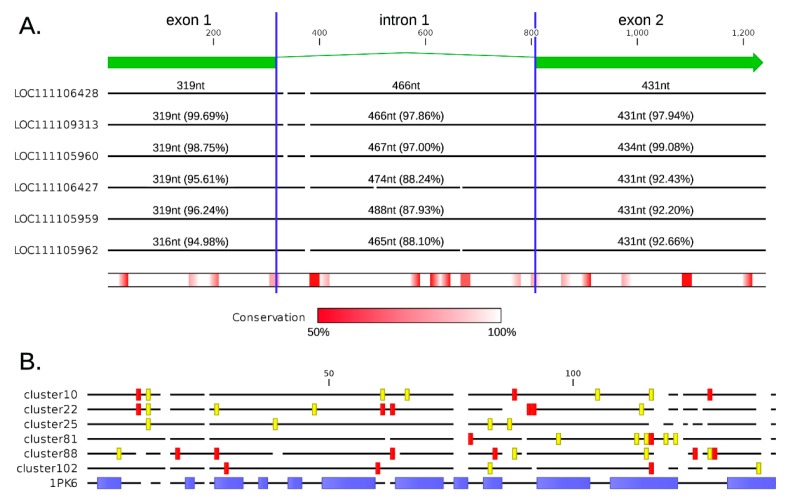
(**A**) Sequence comparison among the six sC1q-like type 2 paralogous genes LOC111105959, LOC111105960, LOC111105962, LOC111106427, LOC111106428 and LOC111109313, all found in chromosome 8. Identity percentages are shown for exon1, intron 1 and exon 2 separately, using the LOC111106428 sequence as a reference. Sequence conservation is shown with a heat map and indels are shown as line breaks. (**B**) Position of sites subject to positive selection in the six largest clusters of paralogous C1qDC genes identified in *C. virginica*. Strongly supported sites (i.e., detected with at least 2 different prediction methods) are shown in red. Sites with moderate support (i.e., detected with just one prediction method) are shown in yellow. The position of the beta strands in the human C1q chain A (PDB: 1PK6) are indicated by blue blocks.

**Table 1 marinedrugs-17-00583-t001:** Details about the number and type of C1qDC genes found in the *Crassostrea virginica* genome.

	chr1	chr2	chr3	chr4	chr5	chr6	chr7	chr8	chr9	chr10	Total
Chromosome size (Mb)	65.67	61.72	77.06	59.69	98.70	51.26	57.83	75.94	104.16	32.65	684.68
Number of C1qDC genes	45	17	18	12	17	36	123	123	73	12	476
C1qDC density (genes/Mb)	0.67	0.28	0.23	0.28	0.17	0.70	2.13	1.63	0.70	0.37	0.70
sghC1q genes	8	0	4	3	1	2	18	63	12	0	111
sC1q-like type II genes	27	6	10	7	12	26	81	36	48	9	262
smultiC1q genes	3	1	0	2	1	0	1	14	0	0	21
sSUEL/C1q	0	10	0	0	0	0	6	0	0	0	16
Other C1qDc genes	7	0	4	0	3	8	17	10	13	3	63

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
