# Peer review of "Extensive Tandem Duplication Events Drive the Expansion of the C1q-Domain-Containing Gene Family in Bivalves"

_marinedrugs, 2019, doi:10.3390/md17100583_

Round 1
Reviewer 1 Report
This study presents a modern genomics level analysis contributing to emerging understanding of the complexities of the innate type immunity in invertebrates, the oyster C. virginica, in this case. Appropriate bioinformatics identify related gene sequences and analyses of spatial distribution and sequence similarity levels are validly interpreted to suggest ongoing mechanisms of gene duplications with selection processes that retain the resutling paralogs.
Explanation of the criteria/message of citation 31 (in lines 198-180) would perhaps better help to clarify the analytical criteria applied. Other minor comments are provided below.
11 suggestS
45-46 chains = monomeric units? complex = multimeric molecule (or system?)
61 suggest edit to
not the "remaining" or "other" two basal clades to indicate complete range of bivalvia
78 guaranteed = effected?
103 suggest edit to
hence belonging to
Also this may the best place to introduce nomenclature of C1q categories, otherwise this remain unclear until methods: e.g 107 hat is a c1q-like type1 protein?
125 what is a QREP?
136 suggest edit to
as "few" as 12
140 suggest edit to
non-homogeneous
164: clarify reciprocal placement? relative position?
171 for clarity suggest edit to
of all 123 C1aDC genes all07
177 targeted = impacted?
232 sugar = saccharide or MAMP?
Author Response
This study presents a modern genomics level analysis contributing to emerging understanding of the complexities of the innate type immunity in invertebrates, the oyster C. virginica, in this case. Appropriate bioinformatics identify related gene sequences and analyses of spatial distribution and sequence similarity levels are validly interpreted to suggest ongoing mechanisms of gene duplications with selection processes that retain the resutling paralogs.
*We thank the reviewer for this positive assessment and his/her constructive comments.
Explanation of the criteria/message of citation 31 (in lines 198-180) would perhaps better help to clarify the analytical criteria applied. Other minor comments are provided below.
11 suggestS
*Fixed
45-46 chains = monomeric units? complex = multimeric molecule (or system?)
*Modified as suggested
61 suggest edit to
not the "remaining" or "other" two basal clades to indicate complete range of bivalvia
*Edited to “remaining”
78 guaranteed = effected?
*Modified as suggested
103 suggest edit to
hence belonging to
*Edited as suggested
Also this may the best place to introduce nomenclature of C1q categories, otherwise this remain unclear until methods: e.g 107 hat is a c1q-like type1 protein?
*Thank you for this suggestion. We agree with this, as we feel it improves readability. Criteria used for nomenclature have been moved here as suggested.
125 what is a QREP?
*We added the complete definition of these proteins from the original publication by Gorbushin, “C1q-related proteins”, aka QREPs
136 suggest edit to
as "few" as 12
*Edited as suggested
140 suggest edit to
non-homogeneous
*Fixed
164: clarify reciprocal placement? relative position?
*Exactly, this has been edited
171 for clarity suggest edit to
of all 123 C1aDC genes all07
*Edited as suggested
177 targeted = impacted
*Modified as suggested
232 sugar = saccharide or MAMP?
*Edited to MAMP, as the referee correctly pointed out we were referring to sites involved in MAMP binding, not necessarily in sugar binding.
Reviewer 2 Report
The manuscript by Gerdol and colleagues describes the genetic diversity and ongoing evolutionary processes that contribute to the architecture of C1q-related genes in a bivalve species.
The manuscript is very interesting and well written. Introduction in particular is an exemplary collection of relevant background explanations and literature review.
I would prefer the text in Results to be more concise. Now there are explanatory passages with interpretation of the results, which would be better situated in the Discussion. Please consider revising
page 2 line 46 also found -> also found in
Figure 1 and Table 1: is there a discrepancy between distribution of sghC1q genes on Chr10?
Please be consistent in nomenclature, i.e. sSUELC1q is not featured in figures, but the category is discussed at length in text.
Caption figure 3 please mention locus at chr 8 again in caption, now it is explained in text only
What is the natural habitat / geographic distribution of the Eastern oyster? Does C1q-gene family diversity reflect the particular niche of this species among bivalves, or are other bivalves under equal ongoing selection pressures?
Are other functionally unrelated gene families also hyperduplicated in the oyster, or is this phenomenon restricted to genes relevant to MAMP recognition? I.e. is the described coarse crossing over examptive or is an unspecific genetic drift observed.
Author Response
The manuscript by Gerdol and colleagues describes the genetic diversity and ongoing evolutionary processes that contribute to the architecture of C1q-related genes in a bivalve species.
The manuscript is very interesting and well written. Introduction in particular is an exemplary collection of relevant background explanations and literature review.
*We thank the reviewer for this positive assessment and his/her constructive comments.
I would prefer the text in Results to be more concise. Now there are explanatory passages with interpretation of the results, which would be better situated in the Discussion. Please consider revising
*The results and discussion sections have been partially modified, as suggested, also by taking into account the suggestion provided by reviewer #1 to anticipate the nomenclature description in the results section.
Specifically, we have moved two explanatory paragraphs from section 2.1 and another one from section 2.3 to the discussion.
page 2 line 46 also found -> also found in
*Fixed
Figure 1 and Table 1: is there a discrepancy between distribution of sghC1q genes on Chr10?
*Thank you for pointing this out! Due to a copy-past error, we had accidentally inverted the numbers of sghC1qDC and sC1q-like type II genes in the table (the figure was correct). We have corrected the table and updated the numbers in the text accordingly.
Please be consistent in nomenclature, i.e. sSUELC1q is not featured in figures, but the category is discussed at length in text.
*Thank you for this suggestion. Table 1 has been modified by breaking up sSUELC1q genes from “others”. In a tentative alternative version of figure 1, we had planned to add another color to mark SUELC1q genes, but we chose to drop this idea as we felt the use of a fifth color would have made the interpretation of this figure more complex.
Caption figure 3 please mention locus at chr 8 again in caption, now it is explained in text only
*Edited as suggested
What is the natural habitat / geographic distribution of the Eastern oyster? Does C1q-gene family diversity reflect the particular niche of this species among bivalves, or are other bivalves under equal ongoing selection pressures?
*The eastern oyster is native to the Eastern Coast of America, covering a very broad range of distribution, from Venezuela to Canada, including the entire Guld of Mexico. As we have previously reported in gerdol et al. 2015, the expansion of the C1qDc gene family does not seem to stem from the adaptation to a particular environment, but it is a common feature of the majority of bivalves, with the exception of the two basal clades, Palaeoheterodonta and Protobranchia. See lines 60-62, where we added a brief sentence mentioning that this gene family expansion is not linked with an adaptation to particular environmental niches.
Are other functionally unrelated gene families also hyperduplicated in the oyster, or is this phenomenon restricted to genes relevant to MAMP recognition? I.e. is the described coarse crossing over examptive or is an unspecific genetic drift observed.
*Although this phenomenon is not restricted to genes relevant for MAMP recognition, several genes involved in immune response, i.e. not necessarily just receptors, but also effectors and those encoding molecules involved in stress response and programmed cell death, are most certainly subject to an increased duplication rate. This subject has been investigated quite in detail by Zhang and colleagues in the Pacific oyster (see https://doi.org/10.1038/srep08693) and, at the transcriptomic level, also in the Eastern oyster by McDowell and colleagues (https://doi.org/10.1016/j.fsi.2016.03.157). The dynamic behind massive gene family expansion and positive selection in bivalves are really quite complex, and unexpected phenomena linked with structural variation might be also somehow involved in explaining the hypervariability of immune-related genes (for further reading see the preprint about the mussel genome, https://doi.org/10.1101/781377). Overall, genomics studies are starting to build some evidence that hyperduplication and hyperdiversification do indeed target immune genes with an increased frequency compared with expectations, but more resequencing data from multiple individuals will be required. We have added a sentence at the end of the discussion to explain that similar evolutionary mechanisms might be expected to underlie the expansion of other immune-related gene families.